# Digitalization of Multi-Object Technological Projecting in Terms of Small Batch Production

**Pavel M. Kuznetsov and Leonid L. Khoroshko \***

Department of System Modeling and Computer Aided Design, Moscow Aviation Institute (National Research University), Volokolamskoe Highway 4, 125993 Moscow, Russia; profpol@rambler.ru

**\*** Correspondence: khoroshko@mati.ru; Tel.: +7-963-753-71-10

**Abstract:** Decreasing the share of time for the technological training of new product creation is an actual and perspective task. The actuality of the given task always increases with the development of machine-building production. Changes which happen inside the organization of industrial production are connected to the development of consumer demand. Modern tendencies are such that there can be seen a constant lowering of batch products with an increase in the range of their production, and, consequently, a reduction in the production stage of the product life cycle. Under these conditions, it is necessary to react as quickly as possible to changes of external (for example, the arrival of new production orders, adjusting the number of already started production, etc.) and internal conditions (for example, stopping technological equipment for organizational reasons, correcting production routes for parts, etc.). Obviously, the effectiveness of the functioning of the production system will be determined primarily by the right decisions. Errors lead to large material costs and a serious loss of time, which can be fatal for an enterprise. The development goal described in the article is to build a digital model of a production system that allows—on a machine time scale and in the absence of material costs—the analysis of various control scenarios and finding correct solutions in in short time. The methodological basis for constructing such models is the use of simulation modeling with the combination of discrete event and agent modeling. The construction of a model on this principle allows its openness to accumulate information about successful decisions made and use them to solve similar problems in the future. The benefit of this approach is the ability to predict the production situation over time, rational distribution of technological resources, reducing equipment downtime, streamlining the routes of production tasks and determining the stages of acquiring the necessary materials and components. It increases the competitiveness of the enterprise and ensures its economic stability.

**Keywords:** automation; modeling; digitalization; production; projecting

## 1. Introduction

In terms of small batch production, the share of time for technological production training significantly grows, lowering the manufacturing performance of new production and its competitiveness at the same time. Taking into account the tendency of lowering the size of new sets of made products and, accordingly, the growth of the product range, the proportion of time spent on technological preparation is constantly increasing. Such a tendency will continue, being a related element of the consumers' demand development. The degree of consumers' demand development characterizes the level of well-being of the country's population and the degree of the satisfaction of industry with new developments in the field of the improvement of technological equipment [1].

The actual problem is task solution for the reduction in the time taken for production technological preparation and the creation of new products, especially small batches, forming a complex of production

tasks. This increases the competitiveness of products through providing an operative reaction to changes in consumer demand and lowering the cost. For achieving this goal, it is necessary to determine a set of methods and means of forming design routes and sequences of project operations and procedures, leading to the achievement of the goal. Methods of building the routes of projecting are defined by the type of project tasks [2].

## 2. Theoretical Basis

In many cases, operative rebuilding in the creation of new products is limited by the abilities of the separately taken production system of the enterprise. At the same time, there is a reserve of technological possibilities, due to the free resources of other similar enterprises. The integration of these resources in one generalized system allows one to significantly widen the technological abilities of a separately taken enterprise [3].

The basis of the developed approach to improving production efficiency is the organization in a single production system of a set of free parts of distributed production systems owned by other enterprises. The strategy for solving this problem is to develop digital models describing distributed production systems, on the one hand, and on the other the aggregation of production orders [4].

The next stage involves the construction of a management process formed by a generalized production system for finding the configuration of the part that most fully meets the requirements for the performance of production tasks. A feature of this management is the ability to make decisions to change its configuration depending on the current production situation. The role of the human operator is reduced mainly to control functions [5].

## 3. Methodology

The methods of solving a given task have a number of procedures: the projecting of technological processes, including the development of route and operational technologies; the generation of possible ways of building technological processes; the verification of generated variants, with the aim of choosing the most successful variants according to different ways of configuring the generalized production system. If necessary, creative approaches include connecting a designer or a group of designers and performing the intellectual part of the solution of the task [6].

All received variants of production system configurations is a virtual production system, because its organization comes to a digital model and is not accompanied by any material restructuring of the existing distributed production systems. This approach provides a fundamentally high flexibility for their structures. As a virtual production system has a rational, sometimes optimal structure, we can solve a whole row of tasks for increasing the effectiveness of completing production tasks:

- fulfillment of production tasks in terms that do not exceed specified ones, but that are close to them;
- minimum costs for the storage of finished products;
- ensuring the minimum cost of production tasks;
- loading of the minimal resources of a generalized production system;
- maximum use of free (idle) units of technological equipment distributed production systems;
- reduction in work in progress;
- significant reduction in the time and complexity of the technological preparation of production.

Thus, the decision of the main given task includes the mobile organization of partly functioning object-oriented production systems for relevant technological processes, which are based on distributed production systems [7].

This approach requires a certain methodological one, because existing optimization methods based on an analytical description of the processes are not applicable for the following reasons:

- complexity of the description of organizational management;

- large dimension of the task;
- non-stationarity of the system;
- high dynamics of interconnections;
- structural complexity of the system;
- complexity of the interpretation of the results.

Based on the above information, simulation modeling was selected as a methodological basis, which allows overcoming these difficulties to a large extent. These models are usually made following the discrete-event principle, which consists of a combination of analytical and logical dependencies. As a result, the functioning of the model provides for discretization in the time of the process of calculating the parameters of the state of the system over time. In the presence of a large number of events in the system and the calculation of the parameters of its state over a sufficiently large period of time, the volume of calculations increases sharply, which greatly complicates the work with this model and requires large computational resources.

Therefore, the following decision was made: to use a combination of a discrete event principle and an agent principle as a methodological basis for solving the problem. The agent one is an object that can perform certain actions in relation to similar objects in its environment, has the ability to perceive part of its environment, and has autonomous behavior.

The agent refers to such structural units of a production system as a production site, line, flexible production system, etc. Dividing the production system into agents allows us to significantly reduce the order of complexity of the system, due to the autonomous consideration of the behavior of the individual system agents. The functioning of individual agents can be represented as follows.

The complex of information flows, formed by information about the production tasks and information about the features of a production system form a generative environment, which is needed for the generation of possible variants of configurations of a production system. A generative environment is a system of entities Object Systems (OS), which can be shown as:

$$OS = (\{sa_i, SA_i\} | i \in N_n\}, \{(sb_j, SB_j) \big| j \in N_m\}),\tag{1}$$

where $sa_i$ and $SA_i$ are, accordingly, the property and its variety of manifestations;
$sb_j$ and $SB_j$ are the base and its variety of elements.

$$N_n = (1, 2, \dots , n) \text{ and } N_m = (1, 2, \dots , m),\tag{2}$$

where n is the number of units of technological equipment in the virtual manufacturing system and m is the number of types of products in the operation assignment.

The complex of relationships can be shown as varieties D1, D2, ..., DN. Then, R is a relation over them, if R is the variety of ordered sequences (n tuples) of the form <d1, d2, ..., dn>, where d1 is an element from D1; d2 is an element from D2; ...; dn is an element from DN. The varieties D1, D2, ..., DN are domains from R.

R consists of a number of tuples corresponding to the number of elements in the production system and represents the power of the many elements that make up a number of possible structures of the virtual production system. The relation can be shown as a table function. Such a view determines the use of relational databases.

Models of the generating medium have a four-level structure:

- model class: tabular, network, permutation, combinative, associative;
- structure of the design decision: single element, ordered set, subgraph, cycle, combination, sample;
- form of interconnection of properties that determine the decision-making procedures: disjunction, conjunction, simultaneous combination of disjunction, conjunction;

- procedure for evaluating the design decision—design decision has a set of properties that is specified in the input data, then there is an equivalence of properties:

$$F(A_i) \equiv F(P_{ik}). \tag{3}$$

Here, there is a set of contours $F$—i.e., a set of structural properties (features). $F(A_i)$ is a set of properties of a design solution, $F(P_{ik})$ is a set of input data properties, and $P_{ik}$ is input data to a $k$-th design object.

If the design decision has a redundant set of properties, the initial set of properties is included in the composition of the design solution properties:

$$F(A_i) \subseteq F(P_{ik}). \tag{4}$$

The binary relations in structural models are described as Boolean matrices.

Relation $\{c_{ij}\}$ represents the $R_A{}^3$ interconnection of elements. $A = \{a_1, \dots, a_n\}$ can be represented as a matrix of relations:

$$\|c_{ij}\| = [A \times A] = \begin{bmatrix} c_{11} & c_{12} & \dots & c_{1n} \\ c_{12} & c_{22} & \dots & c_{2n} \\ \dots & \dots & \dots & \dots \\ c_{n1} & c_{n2} & \dots & c_{nn} \end{bmatrix}, \tag{5}$$

where $c_{ij} = 1$ if the interconnection of elements $a_i$ and $a_j$ exists, and $c_{ij} = 0$ if not. In $[A \times A]$, the first character corresponds to many rows, and the second one corresponds to many columns of a Boolean matrix $\{c_{ij}\}$. Each i-th row (column) of the matrix is considered as a representation of a binary relation $R(a_i)$ between $a_i$ and elements $(a_{i1}, a_{i2}, \dots, a_{i(n-1)})$, for which the elements of the row (column) of the matrix are equal to unity.

The binary relations between the contours of an object *Raf* are described in a similar way. The matrix of the composition of properties of a mathematical model consists of rows corresponding to relations of the form:

$$R_{af} = T(a, F_a), \tag{6}$$

where T is the relation table, $F_a$ is the subset of $F(Fa \subset F)$, and $a$ represents matrix elements.

Binary relations *Raf* are described as Boolean matrices:

$$\|c_{ij}\| = [A \times F(A)] = \begin{matrix} F_1 & F_2 & \dots & F_m \\ \begin{bmatrix} c_{11} & c_{12} & \dots & c_{1m} \\ c_{12} & c_{22} & \dots & c_{2m} \\ \dots & \dots & \dots & \dots \\ c_{n1} & c_{n2} & \dots & c_{nm} \end{bmatrix} & \begin{matrix} a_1 \\ a_2 \\ \dots \\ a_n \end{matrix} \end{matrix}, \tag{7}$$

where $c_{ij} = 1$ if $F_j$ is a part of $F(a_i)$ of element outlines.

The contour matrix of the structural model is disjunctive if the truth values of the contour $F_i$ of the operator acting on the design object are: $F_i(a_k) = 1$, if the circuit $F_i(A_i)$ will be implemented under the influence of this operator, and $F_i(a_k) = 0$ if not—i.e.:

$$F_i(a_k) = \begin{cases} 1, & \text{if the introduction of } a_k \text{ in } A_i \text{ leads to the realization of } F_i(A_i), \\ 0 & \text{if not.} \end{cases} \tag{8}$$

Here, $Fi(ak)$ is a set of properties for the design options for a *k-th* design object, and $k$ indicates the number of a design object's properties in an outline.

In the conjunctive form of communication in the initial and final state of the design object, the truth values of the logical value of the circuit $F_j(A_i)$ to be sold are taken as equal to one $F_j(A_i) = 1$.

The truth values of the circuit - $F_i(a_k)$ of the element $a_k$, including in the $A_i$, are equal to:

$$F_i(a_k) = \begin{cases} 1, \text{ if the element } a_k \text{ participates in the realization of } F_i(A_i) \text{ or does not affect its realization,} \\ 0 \text{ if not.} \end{cases} \tag{9}$$

The matrix of the contours of the structural model in this case is conjunctive.

On the basis of the above-mentioned procedures of choosing and projecting and finding the meaning of attributes, there is a selection of necessary information from the database. According to the demands and restrictions of a higher level, there is a decision about completing a given volume of production tasks. The decision is made based on an analysis of possible generated structures of technological process for completing every production task. The structure of a technological process comes as a complex of routes, descriptions, and results of the selection of equipment and agile tooling, etc. [8].

The process of planning the execution of a production task involves determining the timing of the launch and completion of each production task; the necessary range of technological equipment; the timing of the acquisition of raw materials, semi-finished products, and components necessary for the implementation of the corresponding technology [9].

The control of making production task provides the coordination of intermediate results with the customer's needs and confirming the correctness of the results. Such an approach minimizes the receiving of broken products accounting for possible changes in the technical specifications by the customer and the correction of possible deviations of product parameters on the early stages of their production [10].

Mathematical models of the processes of the interaction of individual agents were based on systemic structural analysis, information theory, varieties, mathematical logic, management, automatized projecting, and programming technology [11]. The verification of a development given results was made in terms of real enterprise with a traditional approach to completing entering production tasks. This provided support for existing production, with the gradual introduction of new approaches [12].

The digitalization of the projecting processes provides the usage of modern tools of computer engineering and the graphic display of information. The model of a multi-object system of technological projecting allows us not only to automate the functions and types of designer activity in automatized production, but also is the base for providing a system approach in projecting [13]. The model includes the "manageable dynamic production" concept, which provides the following consecutive stages: the assessment of the incoming production task, technological design, verification, decision making, and monitoring the implementation of the production task.

The realization of the developed models is based on the widespread usage of computer technology, ranging from individual flexible manufacturing modules, which, as a rule, have built-in microprocessors, and ending with the workstations of designers, technologists, dispatchers, etc. Due to the physical distribution of the mentioned components, the task of the creation of a computer network, which provides cooperation, inevitably comes up [14].

The procedure of generating the possible variants was built on the base of evolution methods, which use genetic algorithms—in particular, the heuristic combining method. The usage of this method allows us to reduce the necessary computing power of computer technology. Every received variant goes under verification. Here, we made an assessment of the current variant for meeting the objecting function conditions. When completing such a requirement, such a variant is said to be workable and there starts its realization in the production system.

If the best received variant of configuration does not meet the given conditions, there is a ranging of the generated population. On the base of this, the forming of a new population is completed, then the process is repeated until we get a working variant.

The process of verification is a complex procedure based on different mathematical approaches, where the main one is the method of imitative modeling. The process of imitative modeling is a summarizing procedure, which is shown as a procedure of searching for local decisions of optimizing processes.

While finding local optimal decisions, methods of linear programming, dynamic programming, and others are used.

The usage of wide scales of optimizing methods widens the possibilities of finding the best variants, speeds up their search, and allows bringing the result to an optimal meaning. Due to this, the system is built on the principle of an open architecture, in which there are added optimizing methods, if they are necessary, which expands its ability to search for the best options [15].

The information support of the projecting processes must include information, which is necessary for searching the best variants of the production system configuration. Information support includes constant, conditionally constant, and variable components. Organizing stored information by this principle allows increasing the reliability of computational processes and reducing the machine time for processing information support.

The constant component of information support has the following types of information:

- world scientific and technical level, fixed in the form of publications, descriptions of patents, and inventions;
- common reference information;
- guiding technical materials developed at the enterprise that define a group of restrictive conditions when selecting and analyzing design options;
- design techniques, which are formalized by the collective experience of specialists in this field;
- archive of accumulated experience in the form of various data on previous developments;
- regulations on document circulation accepted at the enterprise.

The base of a conditionally constant component of information support is formed by:

- classifiers of typical and complex parts;
- technological documentation on typical and group technological processes at this enterprise;
- summaries of typical solutions used in the design of technological processes at a given enterprise;
- reference and regulatory materials;
- forms of output documentation.

The base of a variable component includes:

- information on the contents of the production task;
- information on the composition of the generalized production system;
- information on the production tasks already launched for execution in the production system and those planned for launch;
- information necessary for the implementation of design processes.

The fulfillment of the database with the needed information is made in the process of projecting technological processes in connection with concrete production terms, in which the successful variants of solving optimization tasks were accepted. Thus, there is formed a library of successful decisions, which is used in later projecting processes, being the base for receiving more successful decisions.

The task of information support is giving the necessary data while determining the possible variants of technological processes and their possible modification, while completing the given production tasks. For solving technological tasks, it is necessary to imagine if there is any pattern of combination of surfaces in the shape of the part or if there is possible classification of typical detail forms on this base.

For studying the character of combining the elementary surfaces in order to typify technological solutions, the most promising is the classification of machine parts according to the functional and structural features. These signs are the criteria for creating various classification systems of machine parts, which are the basis for the development of typical technological processes for the manufacturing of similar parts. The whole variety of machine parts comes down to a limited number of classes—for example, the class of shafts, of bushings, of body parts, etc.

While, in the mentioned classifications, the detail form is one of the main factors of separation in classification signs, we can think that such an approach is rather right for reflecting the typical for each unit connection of the main surfaces' forms. For example, for the class of shafts, there is the combination of external concentric surfaces of revolution will be predominant and typical; for the class of bushings, there is the combination of external and internal surfaces of rotation; for the class of body parts, there is the combination of various kinds of planes with internal surfaces of revolution, etc. Such an approach allows determining the possibilities of changeovers of technological equipment for making other technological processes, while changing the details' nomenclature.

The forming of the detail from a homogeneous work piece depends on manufacturing methods that provide the formation of the form of an elementary surface, of the form of the combination of elementary surfaces, and of the shape of the part as a whole. Thus, there will be the following ways of creating the detail's form [16].

The first way is the usage of methods which have the least technological abilities; during one operation, we can create only one form of an elementary surface $F_{surf}$. In this case, the form of the detail will be created due to the consistent creation of elementary surfaces' forms:

$$F_{surf_1} \cup F_{surf_2} \cup ...F_{surf_i} \cup ... \cup F_{surf_n} \rightarrow F_D. \tag{10}$$

$n$ shows the number of surfaces that define the geometric shape of the part.

The second way is usage of methods which provide the creation of form of combining elementary surfaces during one operation. In this case, the detail's form will be created by the consistent creation of combining elementary surfaces:

$$F_{comb_1} \cup F_{comb_2} \cup ...F_{comb_j} \cup ... \cup F_{comb_n} \rightarrow F_D. \tag{11}$$

The third way is the usage of methods which provide the creation of elementary surfaces and their combinations during one operation. In this case, the form of the detail will be created through the consistent creation of elementary surfaces and their combinations:

$$F_{surf_1} \cup ... \cup F_{surf_M} \cup ...F_{comb_1} \cup ... \cup F_{comb_L} \rightarrow F_D. \tag{12}$$

The fourth way is the usage of methods which provide the creation of a whole details' form during one operation:

$$F_{\sum surf_i} \rightarrow F_D. \tag{13}$$

The forming of the detail's form through the connection of different parts is used during making oversized details, when the energetic power and overall size of the equipment limit the possibility of the formation of forms; in the manufacturing of parts, the shape of which consists of a combination of elementary surfaces, it is technologically difficult to implement in a single work piece. In the last case, the detail is separated into different parts, the forming of which does not cause technological difficulties. The creation of the form of different parts will obey the laws of formation of the shape of the part from a single work piece. The connection of parts into a single part is integral (welding, soldering, etc.).

## 4. Results

The digitalization of mathematical modeling processes based on simulation using a combination of discrete event and agent principles has allowed building a digital analogue of a production system that reflects the qualitative and quantitative characteristics of a real production system. A positive difference of this approach is the versatility and openness for making changes when adapting the model to production systems with other structures and parameters.

The described work results were obtained using the developed proprietary software (Tsyrkov, G.A.; Tsarkov, A.V. Software platform for the integration of automated systems. Server part. 2020615161,

18 May 2020). The machines were used in 1 (turning), 2 (drilling and boring), 6 (milling) and 7 (planing, slotting and broaching) groups according to the Experimental Research Institute of Metal-Cutting Machine Tools classification (Russia). This is available online: https://pellai.com/tehnicheskie-discipliny/spravochnaya-informaciya/klassifikaciya-metallorezhushhih-stankov/ (accessed on 23 July 2020) [17]. The standard tool used on these machines has no special features.

During the simulation, the following results were obtained or confirmed.

The choice of the degree of processing operations' concentration is fundamental in multi-tool processing. With the increasing of the amount of tools in adjustment, productivity grows to a certain limit. The further increasing of their amount can lower the productivity in connection to the growth of time spent on changing and adjusting tools and lower cutting speeds. In addition, the cost of processing increases, since more complex adjustments are carried out on more expensive equipment and involve the usage of more complicated technological snap.

The rational combining of technological transitions define, depending on mutual the location of the machined surfaces, the possible placement of cutting tools in the working area and unhindered removal of the resulting chips. The insufficient rigidity of the harvesting often stops the parallel execution of technological processing transitions. High accuracy classes are separated from previous finishing using single single-tool (or low-tool) sequential processing schemes.

Important meaning in clarifying the structure of the processing operation on multioperational machines and establishing the degree of their concentration has a factor in the reliability of the machine. With an increase in the degree of concentration, the number of cutting tools in adjustments increases, the technological equipment and the design of the machine become more complicated, and the reliability of its operation decreases. With increasing of the number of failures, the total time to resolve them during the shift, and, therefore, the total production time of a batch of parts increases [18].

The disadvantage of typical technological processes is a weak loading a part of the equipment fleet; therefore, in conditions of small-scale and mass production, it is advisable to apply easily adaptable equipment based on computer numerical control machines. In the absence of this enterprise equipment, it is necessary to expand the technological capabilities of existing machines, applying, e.g., devices such as multi-spindle drill heads with adjustable spindles, three- or four-spindle milling heads, etc. On these machines with a small readjustment, it is possible to process different parts with structural differences and included in one type. For a full load of equipment in the lines, it is advisable to create group operations on unloaded machines, processing parts of several types, or even classes. These operations will be common to several lines of processing different types of parts. That is why, for the maximum usage of typical and group technological processes and the possibility of creation closed-loop production lines, we should use an integrated method of production preparation based on the simultaneous use of standard and group technological processes, the development of standard processes and group operations common to several types parts, and the creation of easily adjustable equipment and devices for standard and group technological processes.

The creation of a database of ready technological processes allows reducing the bunch of irrational processes, which are used to the optimal number. Typing of technological processes based on digitalization of production processes is the basis preparation of production based on a synthesis of the experience of the design team technological processes in order to create information materials to determine the most productive, progressive technological processes for manufacturing parts on a specific the enterprise, taking into account the existing fleet of technological equipment, agile tooling, tool base, etc.

## 5. Discussion

The results, which were received in the end of the research, are confirmed due to the developed methods for monitoring the condition of equipment of flexible manufacturing systems of others authors [19]. The usage of developed databases of accumulated successful solutions allows using innovative approaches to expand the technological capabilities of equipment for production systems [18].

The obtained approaches can serve to integrate private technological solutions, aimed at improving the efficiency of the operation of certain types of equipment of a production system [20]. Thus, the described system acquires the properties of an open one, allowing it to build up during its operation with new solutions useful for increasing its efficiency [15,21].

## 6. Conclusions

The digitalization of the processes of multi-object technological projecting in terms of small production allows making a virtual model of a production enterprise's system, which provides a variable process of searching rational decision of organizing production processes of making the products of machine-building industry on the stages of technological production training. The given approach based on the formation of a generative information environment allows predicting the results of production decisions on a machine scale time, and by analyzing the results of modeling we can choose the best one in accordance with the current production situation.

The lack of a material component while modeling the processes of production training provides minimal costs in making experimental studies, while searching for rational decisions with minimum working hours.

The usage of the proposed development allows organizing production processes with the maximum load of existing technological equipment, significantly increasing the productivity and reducing the cost of manufacturing.

Perspective modeling and the formation of a database of successful production solutions allows reducing the time of the technological preparation of new product manufacturing.

**Author Contributions:** Conceptualization, L.L.K.; Methodology, P.M.K.; Formal analysis, L.L.K.; Investigation, L.L.K. and P.M.K.; Resources, P.M.K.; Data curation, P.M.K.; Writing—original draft preparation, L.L.K.; Writing—review and editing, P.M.K. and L.L.K.; Project administration, L.L.K. All authors have read and agreed to the published version of the manuscript.

**Funding:** This research received no external funding.

**Conflicts of Interest:** The authors declare no conflict of interest.

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
