# Peer review of "Digitalization of Multi-Object Technological Projecting in Terms of Small Batch Production"

_inventions, doi:10.3390/inventions5030038_

Round 1
Reviewer 1 Report
Manuscript titled Digitalization of multi-object technological projecting in terms of small batch production was clearly written. A very interesting research and excellent manuscript.
Few comments only
Abstract = can be improved
Introduction
The introduction is concise and up to the point.
Materials san Methods
Well-described, with sufficient details to enable other research groups to repeat similar experiments.
Results and discussion
Excellent report of the findings and their explanation. No comments
Author Response
Dear sirs,
The authors reviewed the abstract once again and made several changes to improve it. Thank you for the review.
Best regards,
Authors
Reviewer 2 Report
Theoretical basis. The motivation for multi-object technological projecting needs further clarification.
Methodology. The mathematical modeling of the problem should be carefully checked, corrected, and improved. There are a lot of errors in the following equations:
- Please check equation in line#121; also pay attention to brackets;
- Please define F(Ai), F(Pik), and Pik – equation in line#143;
- What exactly an represents (line#148)? Please explain.
- RAF or Raf (lines#155-160)? Please be consistent in the whole text;
- Please define and explain a (line#158);
- F(Fa ⊂ Fa), line#159?
- Please define Fi(ak), lines#164-166; what does the k represent?
- Please check again the equation in line#169. What does Fj(ak) represent?
- What does n represent (equations in lines 282 and 286)?
- What does Σ???ℎ? represent (equation in line#293)?
Numeration of equations should be provided.
Results. This part of the manuscript should be significantly improved. Please explain the settings and conditions of the experiments. Provide information about machine tools, cutting tools, manufacturing equipment, technological processes, relevant databases, etc. Explain parameter settings and mention software or program package used for simulation. Please show, analyze, and discuss the experimental results (figures, tables, diagrams, etc).
Language. The manuscript should be rewritten and thoroughly revised by an English language editor to eliminate numerous grammatical and logical mistakes. For example, the following sentences should be carefully rewritten/corrected:
Line#41: "Taking into consideration the tendency of lowering the sizes of new made products’ batches, and accordingly with the growth of the product range, the proportion of time spent on technological training is constantly increasing."
Line#50: "This increases the competitiveness of products through providing an operative reaction on changes of consumers demand and lowering the self-cost of the creation."
Line#58: "Integration of this resources in one generalized system allows significantly widen technological abilities of a separately taken enterprise [3]."
Line#71: "Methods of solving a given task provides completing a row of procedures: projecting technological processes, including development of route and operational technologies;..."
Line#102: "These models are usually built on a 102 discrete-event principle, which consists in a combination of analytical and logical dependencies. "
Line#174: “Coming to the decision is made based on a analysis of possible generated structure of technological process for completing every production task.”
Line#228: “In the base of a constant component of information support there are…”
Line#277: “Thus, there will be the fllowing…”
Author Response
Dear sirs,
Thank you for the review. Please find attached the document with our response.
Best regards,
Authors

Round 2
Reviewer 2 Report
The authors improved the manuscript according to suggestions.